# Electric-Field Mapping of Optically Perturbed CdTe Radiation Detectors

**DOI:** 10.3390/s23104795

**Published:** 2023-05-16

**Authors:** Adriano Cola, Lorenzo Dominici, Antonio Valletta

**Affiliations:** 1Institute for Microelectronics and Microsystems, IMM-CNR, Via Monteroni, 73100 Lecce, Italy; 2Institute of Nanotechnology, NANOTEC-CNR, Via Monteroni, 73100 Lecce, Italy; lorenzo.dominici@nanotec.cnr.it; 3Institute for Microelectronics and Microsystems, IMM-CNR, Via Del Fosso Del Cavaliere, 100, 00133 Rome, Italy; antonio.valletta@artov.imm.cnr.it

**Keywords:** radiation detectors, electric field imaging, Pockels effect, polarization, deep levels, semi-insulating, CdTe

## Abstract

In radiation detectors, the spatial distribution of the electric field plays a fundamental role in their operation. Access to this field distribution is of strategic importance, especially when investigating the perturbing effects induced by incident radiation. For example, one dangerous effect that prevents their proper operation is the accumulation of internal space charge. Here, we probe the two-dimensional electric field in a Schottky CdTe detector using the Pockels effect and report on its local perturbation after exposure to an optical beam at the anode electrode. Our electro-optical imaging setup, together with a custom processing routine, allows the extraction of the electric-field vector maps and their dynamics during a voltage bias-optical exposure sequence. The results are in agreement with numerical simulations, allowing us to confirm a two-level model based on a dominant deep level. Such a simple model is indeed able to fully account for both the temporal and spatial dynamics of the perturbed electric field. This approach thus allows a deeper understanding of the main mechanisms affecting the non-equilibrium electric-field distribution in CdTe Schottky detectors, such as those leading to polarization. In the future, it could also be used to predict and improve the performance of planar or electrode-segmented detectors.

## 1. Introduction

The electric field is the main driving force for the separation and collection of the charges generated inside radiation detectors (X-rays and γ-rays) based on semi-insulating materials. Its internal spatial distribution is therefore crucial for its correct operation, where a uniform and stable profile provides ideal working conditions. In real detectors, an unavoidable internal space charge is present, but this is acceptable as long as the electric field is distributed throughout the detector without inactive regions. Continued exposure to radiation can perturb the space charge and thus the electric field. This perturbation affects the fixed charge by trapping and emission processes involving both donor and acceptor levels inside the detector material. Polarization is often observed at high X-ray fluxes, with detrimental effects on the performance of Cd(Zn)Te detectors. Such radiation-induced polarization is the typical signature of a space charge build-up [1], mainly related to hole trapping. In the dark, polarization can also be observed in diode-like CdTe detectors (bias-induced polarization). In this case, the space charge build-up is related to hole emission from the carrier-depleted region [2,3] in voltage-biased detectors. Over time, the depleted region slowly reduces its extent, with the consequent squeezing of the electric field under the blocking contact.

In the literature, the study of electric-field perturbation has mainly been carried out by irradiating the device with optical photons [4,5,6,7,8,9,10,11], with energies above and below the energy gap. For photons with energy below the energy gap, the optical transitions between localized and extended states directly change the charge state of the localized states, but this is a weak effect due to the low absorption coefficient. In this case, it has been shown that space-charge manipulation by optical photons at specific wavelengths during X-ray exposure can be beneficial, allowing for improvements in the X-ray detection [8,9,12,13,14]. For photons with energy above the bandgap, space-charge manipulation is indirect, mediated by free charge generation, but greatly enhanced. In this case, a large number of carriers are photogenerated in a short absorption length before being trapped and detrapped [6,7,15]. Obviously, similar mechanisms affect the electron-hole pairs photogenerated by the X-rays, but in this case—due to the different interaction mechanism—a much longer effective absorption length results. The main advantage of using optical photons over X-rays is that it simplifies the experimental setup and provides greater flexibility in perturbing the space charge.

Often these studies rely on the Pockels effect, the birefringence induced by the electric field itself in non-centrosymmetric crystals such as Cd(Zn)Te. This electro-optical effect has been shown in recent years to be a convenient tool to probe the internal electric-field distribution both in the dark [2,16] and under external radiation [7,10]. However, these studies have been mainly limited to one dimension, i.e., probing the field profile along the direction perpendicular to the device, especially for large planar electrodes. On the other hand, access to the electric field along other directions or even its 2D mapping could be particularly valuable. For example, there are issues for segmented electrodes regarding charge sharing or collecting loss in low field regions between adjacent electrodes [17,18]. Exposure to spatially localized and intense radiation fluxes can lead to strong lateral perturbations, which need to be qualified as they can greatly affect the performance of nearby pixels [19,20,21]. In addition, the presence of material inhomogeneities [22,23,24] and spatial defects is a major problem in Cd(Zn)Te devices. Their effect on the detection performance and ultimately on the radiation-induced polarization has been revealed by means of carrier transport maps, a rather demanding technique based on scanning highly collimated X-ray synchrotron beams [25,26]. This is another example where electric-field mapping could prove to be a very useful diagnostic tool.

A recent study on two-dimensional electric-field mapping was published by Dědič [27], where an upper strip electrode was used to induce transverse field inhomogeneity. The reconstruction of the electric-field vector in the plane perpendicular to the strip was achieved by performing several independent Pockels measurements. We have recently shown [11] that an accurate reconstruction of the electric-field vector in two dimensions is possible even from a single electro-optical image when using planar electrodes, by exploiting the conservative property of the field in the electrostatic case. In this case, transverse inhomogeneity was obtained by using a cylindrical lens to focus the exposure beam into a linear optical perturbation on the cathode side. This allowed us to retrieve the electric-field distribution in the transverse plane and follow its time evolution. Here, we use the same experimental approach. However, in this case, we direct the perturbing beam onto the anode side of the planar Schottky CdTe:Cl radiation detectors. Therefore, the above optical photons are strongly absorbed in a region where the unperturbed field is moderate to high, unlike the case analyzed in [11] where they arrive in the low-field region. We can thus see a major difference in the effect of the optical perturbation: in the previous case [11], the field is suppressed in a large region under the cathode and squeezed the field in a limited region at the anode; in contrast, in the current configuration, the field is balanced in the central region, making it more homogeneous there and extending it to the cathode. This could have implications for the performance design of the detectors. In addition, the current work provides a further validation of the previously discussed compact two-level model in Schottky CdTe detectors. Indeed, we show that the model is able to fully predict the spatial and dynamic behavior of the electric field in two dimensions after the application of voltage and optical bias. To this end, numerical simulations are performed, and the results are compared with experiments.

## 2. Experimental Methods and Numerical Simulations

The experiments were carried out on an Acrorad CdTe:Cl Schottky detector with dimensions of 4 mm × 4 mm × 1 mm. Two planar electrodes embed the CdTe crystal on the square faces: a hole-blocking indium and a quasi-ohmic platinum contact (In/CdTe/Pt). The semi-insulating CdTe crystal is weakly p-type and (111)-oriented. The collimated probe beam (continuous wave, 980 nm) has a diagonal linear polarization (45° from the vertical) adjusted by a polarizer. It hits the lateral surface of the sample and undergoes the Pockels effect as it passes through the detector. At the exit, its perpendicular (−45°) polarization component is analyzed by another polarizer and sent to an NIR-enhanced camera equipped with a zoom lens. The perturbing light beam (pulsed wave, see below) impinges vertically on the device. It is focused by a cylindrical lens into a 150 μm-wide and >4 mm-long line on the upper anode contact. The direction of such a line is parallel to the direction of propagation of the probe beam, which arrives laterally and is transmitted across the entire detector (see a sketch of the experiment in Figure 1). Such a geometry ensures uniformity of the electric field along the direction of the probe beam. The sample is mounted on a thermal stage and kept at 40 °C, and its position is controlled along six degrees of freedom.

The optical perturbation beam comes from a supercontinuum laser, i.e., a pulsed laser with a broad white spectrum, filtered by a long-pass filter with a cut-on wavelength of 780 nm, resulting in an optical power of 23 mW. Such a laser has a repetition rate of 24 kHz with a pulse width of 1–2 ns. Here, we are interested in the long-term dynamics of the space charge, so the pulsed nature of the beam is not relevant. Basically, we integrate over several pulses and do not detect the faster dynamics associated with free carriers. The measurement procedure involves the acquisition of successive images, with an exposure time of approximately 100 ms and a sampling time of 2 s, during different stages. These stages include the initial forward voltage sweep in six steps of 100 V: the central phase in which the detector is kept biased at 600 V, first in the dark (5 min) and then optically irradiated by the perturbing beam (5 min); and then again in the dark (>15 min), followed by the final backward voltage steps to zero bias.

Two-dimensional numerical simulations were performed using the semiconductor device simulator “Sentaurus”, part of the Technology CAD software package provided by Synopsys, Inc. [28]. We have used the same approach and model [11] that has proved effective for cathode-side optical irradiation. The simulations are based on the implementation of both the device and the optical interaction modelling. The basic parameters of the device are the In and Pt electron barrier heights (Φ_In_ = 0.5 eV, Φ_Pt_ = 0.8 eV), which define the hole-blocking and quasi-ohmic (slightly hole-injecting) contacts. The simulated 4 mm × 1 mm CdTe is mainly described by a two-level model, a fully ionized shallow donor and a deep acceptor. The latter acts as a compensation layer, resulting in the slightly p-type semi-insulating material. The energy of the acceptor level is fixed at E_a_ = 0.725 eV from the valence band.

Other parameters, including the optical irradiance of the perturbing light, were varied to find the ranges that gave reasonable agreement with the experimental results. Appendix A lists the relevant parameters and the values used in the present simulations. Furthermore, following the experiments, an incident optical beam of 150 µm in width on the anode side has been considered, also taking into account the spectral absorption [29] of the CdTe crystal. In the simulation run, the entire experimental voltage/optical bias sequence was emulated.

## 3. Results

### 3.1. Electro-Optic Images and Electric-Field Maps

The original intensity maps obtained in the crossed polarizer transmission configuration are shown in Figure 2. These maps are normalized using the image taken with parallel polarizers (representing our reference transmission). The images are acquired at selected times: at 600 V voltage bias with no irradiation (Figure 2a), after 5 min under optical irradiation (Figure 2b), then 2 s (Figure 2c) after switching off the light, and 15 min (Figure 2d) later, under dark and still under the 600 V voltage bias. For our geometry, from a digitized Pockels image P(x,y) and from the maximum intensity P_para_(x,y), acquired with parallel polarizers (both at 45°) and no applied voltage, the electric-field map is retrieved by inverting the electro-optic relation:(1)$$P\left(x,y\right)/P_{\mathrm{para}}\left(x,y\right)=\sin^{2}\left[\frac{\pi }{2}\frac{E_{y}\left(x,y\right)}{E_{o}}\;\right]$$
where E_o_ is the field value at which maximum transmission is observed (i.e., half-wave retardation between horizontal and vertical linear polarizations). Figure 2e–h show the field extracted from the previous intensity maps using a custom fitting procedure. The procedure consists of fitting each vertical profile by using a heuristic form for the underlying E_y_(y) and applying Equation (1) to it. We note that Equation (1) also reflects the fact that for the symmetry class of our crystal and its orientation, only the vertical component of the electric field E_y_(x,y) can be inferred when using crossed polarizers at −45° and 45° with respect to the vertical axis [27]. For the unperturbed case in Figure 2a, no E_x_ components are expected, since the planar electrodes induce a laterally homogeneous potential, i.e., its gradient is only along the device depth. This is associated with a purely vertical electric field E_y_(x,y) = E(x,y), as shown in Figure 2e. For the cases of Figure 2b–d, we evaluate the horizontal component E_x_(x,y) by applying the conservative feature of the electrostatic field [11], thus finally obtaining the full vector E(x,y). Streamlines on the maps visualize the directions and intensity of the electric field, showing the effect of the optical perturbation at the end of the irradiation (Figure 2f). It can be seen that the perturbation increases laterally with increasing distance from the irradiated anode plane. It can also be seen that the overall shape of the field amplitude shows a saddle point near the central position. The overall perturbation persists and is almost constant 2 s after the irradiation is turned off (Figure 2g), while it undergoes only a slow recovery in the following period. The perturbation is reversible and becomes almost negligible 15 min after the light is turned off (Figure 2h).

The central vertical sections of the field corresponding to the maps are shown in the right panels of Figure 2, with y increasing from the anode to the cathode. Prior to irradiation, the field follows a linear decrease starting from the anode, as shown in Figure 2i, while the profiles are uniform along the x-direction. The linear decay is due to the hole-blocking behavior of the anode contact. The effect of the optical irradiation is to induce a convex shape in the vertical profiles (Figure 2j), with a significant increase in the field near the cathode. In fact, the electric field tends to generally decrease in the region under the optically irradiated anode, and it is compensated for by the increase on the opposite side.

As can be seen from a comparison of the maps in Figure 2f,g with the corresponding profiles in Figure 2j,k, there is little difference after the irradiation switch-off (2 s later), when all of the photogenerated free carriers are already collected (or trapped).

This does not mean that the free carriers do not perturb the electric field during their transit time. In fact, their remarkable effect will be the subject of a forthcoming paper. In the present experiments, their effect is simply not visible in the Pockels images, considering that they are recorded with a long exposure time (about 100 ms) and the fact that the time between successive optical pulses (about 40 µs) is very long compared to the hole collection time (about 0.2 µs). Under such experimental conditions, the relatively fast effect of the free carriers on the electric field becomes negligible when integrated in the Pockels images.

From the spatial profiles in Figure 2, it can be seen that the electric field has a negative and almost uniform slope throughout the detector under dark conditions. The slope becomes positive near the cathode under irradiation. The last profile in Figure 2l shows that 15 min after the optical irradiation, the field has almost restored its linear behavior to that seen before irradiation but with a larger slope because some bias-induced-polarization has accumulated.

The presence of opposite slopes in the electric field indicates that both positive and negative space charges are simultaneously present in the detector, at different locations. This is peculiar to compensated semi-insulating materials, where the equilibrium Fermi level in the bulk remains close to the energy of a dominant deep level, which is thus characterized by partial charge occupation. Consequently, the net space charge in a given region of the detector results from an increase or decrease in the occupation of the compensating deep level.

In addition, if the electron and hole-capture cross sections are comparable, which is the case for energy levels close to the mid-gap, then such levels can communicate with both extended bands, thus easing the possibility of a net positive or negative space charge.

Deep-level occupancy changes naturally occur near interfaces with metal electrodes and are then affected by the current transport mechanisms across such interfaces. A clear signature of this is provided by the behavior of the quasi-ohmic Pt/CdTe/Pt detectors fabricated using the same CdTe bulk material as in the present work. In fact, Pt is a hole-injecting contact in CdTe [30], which causes the presence of a net positive space charge in the biased detector. Therefore, in contrast to Schottky In/CdTe/Pt detectors, in quasi-ohmic Pt/CdTe/Pt detectors, the electric field drops when moving away from the cathode side [31].

### 3.2. Space-Charge Retrieval

A further perception of the optical perturbation can also be seen by mapping the space charge. Initially, the charge is basically uniform and negative at pre-irradiation levels of about Q_sc_ = −0.66 × 10^12^ cm^−3^. This was obtained from the Poisson equation in the case of Figure 2e,i. During exposure, the negative charge around the irradiated anode decreases strongly, reaching a minima around −4.5 × 10^12^ cm^−3^, while the space charge becomes positive in a large region around the cathode, as shown in Figure 3, which refers to the time at the end of the optical irradiation. The top panel, Figure 3a, shows the horizontal profile of the charge near the cathode. The superimposed line (solid cyan) shows a Gaussian fit in very good agreement with the points (blue dots) obtained from the original image. The baseline of the fit is negative, close to the unperturbed previous value, while the central part assumes positive values. The FWHM of the fitting Gaussian is about 1 mm, the area of the charge inversion is even wider laterally. The boundary of the charge inversion is clearly visible in the full 2D charge map in the bottom panel, Figure 3b, where the white color marks the Q_sc_ = 0 region. It can be seen that the space charge becomes positive over a region whose lateral size gradually increases with depth from the anode side. As mentioned, it is larger than the 1-mm detector thickness near the cathode.

### 3.3. Numerical Profiles at Increasing Power

It is now interesting to compare the experimental profile of the most perturbed situation, that which corresponds to the end of irradiation, Figure 2j, with those obtained from the numerical simulations. The simulations reproduce the whole bias/light experimental procedure, including the exposure time of 5 min of optical irradiation as in the experiments. We have performed simulations with different optical powers, resulting in the field profiles shown in Figure 4. A progressive decrease (increase) of the electric field near the anode (cathode) can be seen, as the optical power is increased. This can be pushed up to a complete inversion of the profile, where its slope becomes positive basically everywhere in the detector. As a matter of fact, the electric field tends to decrease in the region under the optically irradiated electrode, reversing its slope almost completely at high optical irradiances, as can be seen in Figure 4.

Previous results [5,6,11] have indicated that for CdTe detectors, this behavior is independent of the specific electric-field profile before irradiation. It is also independent of the type of irradiated contact, i.e., blocking or ohmic, and of the bias sign, i.e., anode or cathode. In agreement with Franc [32], we can state that at high optical irradiances, the influence of the contacts becomes less pronounced with respect to the suppression induced by the optical irradiation. In addition to device-dependent peculiarities, electric-field suppression under the optically irradiated contact is expected due to the continuity of the electric current.

In terms of comparison with the experimental results, the best matching profile is that labeled 64 in Figure 4, in good agreement with the experimental profile of Figure 2j. This also corresponds to the maximum optical irradiance possible in our setup. We note that in the experiments, despite the appreciable incident optical power, the perturbation is limited (and a complete slope inversion has not taken place). This may be due to the high optical opacity of the anode contact, which limits the effective power reaching the underlying CdTe.

Concerning the bulk model used in the two-dimensional numerical simulations, we have adopted here the same two-level model (a shallow donor and a deep acceptor) used in a previous work [11], where similar experiments were performed on the same detector, as well as their simulations, but irradiating from the cathode side. Remarkably, almost identical input parameters reproduce the main experimental features quite well, both in the case of cathode irradiation and in the present case of anode irradiation. For a more detailed comparison of the spatial distributions of the electric field, see Appendix A with the simulated maps and profiles corresponding to the experiments shown in Figure 2.

We note that the effect of intermediate levels of optical irradiance on the anode side certainly improves the electric-field uniformity. However, this is not a solution for the polarization effects, since the high photocurrent levels under the optical irradiation are incompatible with the simultaneous acquisition of X-ray signals. In addition, the field uniformity is not maintained after switching off the optical irradiation. On the contrary, as seen in the discussion, the global effect of the irradiation is to accelerate the negative space charge build-up, i.e., the bias-induced polarization.

### 3.4. Time Dynamics

The associated time dynamics of the field induced by the optical perturbation beam are shown in Figure 5a. Here, we show the evolution of the electric field at two specific points of the detector located at the horizontal center of the device, and thus of the optical beam, at a vertical distance of 50 µm from the anode or cathode plane, respectively. The initial steps correspond to increasing the voltage in steps of 100 V up to 600 V in the dark. Each step immediately increases the field at both the anode and cathode, but is followed by slow transients during which the electric field tends to increase further at the anode and decrease back at the cathode. During the 5 min of optical irradiation, the electric field reverses such a trend: it slightly decreases at the anode, while a strong increase towards saturation is observed at the cathode. This is due to the capture of the free holes drifting from the anode throughout the detector, the most striking consequence of which is the addition of a positive space charge near the cathode, where it was negligible in the dark.

After the light is turned off, the free carriers are quickly swept out, the hole emission becomes the dominant process everywhere, and bias-induced polarization starts again with its slow characteristic time constant.

The last steps of the field at the anode correspond to the voltage sweep back to 0 V. For the sake of comparison, Figure 5b shows the corresponding transients from the simulations. As for the spatial distributions, the temporal evolution is well reproduced, both qualitatively and quantitatively, for the entire sequence of the experiment. It is noteworthy that the observed transients are strictly related to both the electron and hole-capture cross sections of the compensating deep level. As a general result, our numerical simulations confirm that these capture cross sections must be very small (10^−18^–10^−19^ cm^2^) and of the same magnitude in order to find good agreement with both the anode and cathode irradiation experiments.

We observe slow changes in the electric field, both during the carrier photogeneration, when hole capture prevails, and in darkness, when hole emission becomes dominant. The former process, hole capture, could be expected to be faster and is limited in our case by the modest amount of photogenerated carriers. As a matter of fact, we observed faster transients when exposing to stronger light. The latter process, hole emission, is naturally slow due to the large energy difference between the deep level and the valence band. Such emissions, which are the underlying mechanism of the bias-induced polarization, dominate both when the voltage is first applied and when the optical perturbation is removed. The associated time constant is independent of the bias and optical irradiation.

### 3.5. Full Time–Space Charts

Time–space charts of the horizontal electric-field profiles during the whole optical irradiation procedure are depicted in Figure 6, highlighting the spatial evolution of the field decrease observed near the anode (Figure 6a) and the field increase observed near the cathode (Figure 6b). The extension of the regions affected by the optical perturbation can be clearly seen in both panels. Near the cathode, during the optical irradiation, the lateral perturbation seems to saturate in the first minute, while at the x-center coordinate, as we have already seen in Figure 5, the field continues to grow slowly. When the light is turned off, the field recovers more quickly as the distance from the x-center increases. Near the anode, an almost complementary behavior can be observed, although the field variations are more limited here. Interestingly, it is possible to observe a memory effect near the x-center that has not yet been appreciated: a brighter band is visible in the central part of the chart, which remains with similar intensity until the end of the recordings under dark conditions, 15 min after the light is turned off. Interestingly, this effect is even more noticeable during the subsequent voltage sweep back to 0 V, but this memory is lost when a subsequent measurement sequence is performed. In order to quantify the lateral extension of the perturbation, we plot in Figure 6c the horizontal profiles of the field (vertical component) during the optical irradiation, extracted from the time charts of panels a,b at selected times. It can be seen that the profiles at the anode and cathode sides are almost complementary, with the lateral extension of the perturbed region being slightly larger at the cathode side.

## 4. Discussion

Our results show that the local free carrier photogeneration can affect the electric-field distribution over a large spatial range (with respect to the excitation region) and over a long time scale (with respect to the typical carrier collection times) in CdTe Schottky detectors. The field perturbation results from the induced changes in the fixed space charge. In compensated semi-insulating materials, such changes involve the deep level responsible for electrical compensation and occur through standard charge capture and emission processes, described by the Shockley–Read–Hall statistics. In the present configuration, the photogenerated holes play a more prominent role with respect to the photogenerated electrons, because the latter are collected quickly. On the other hand, the holes generated near the anode are trapped by the compensation level, while they drift towards the cathode and eventually build up a positive charge there. During optical irradiation, or at least at its beginning, hole trapping near the cathode is more effective, because the holes experience lower and lower electric fields as they drift. This leads to an accumulation of positive space charge near the cathode, which is counterbalanced by the negative charge near the anode, resulting in the observed electric-field convexity.

Interestingly, the enhanced negative space charge near the anode continues to increase even after the optical excitation is removed, due to the hole-emission process. In fact, since hole emission is the dominant process, the space charge rigidly shifts to more negative values throughout the active region. This phenomenology is evident when looking at the simulated profiles of the space charge in Appendix A, which explains why the electric-field profile does not recover the linear shape even 15 min after the light is turned off (Figure 2l).

It is worth noting that the occupancy of the acceptor level is also central in the opposite case, i.e., for optical irradiation on the cathode side. As shown before, the strong increase in the negative space charge can be attributed to the capture process of electrons generated at the cathode side and drifting to the opposite side. This two-fold behavior is possible because, as we have seen, the capture cross sections for electrons and holes are comparable, as is usually the case for energy levels close to the mid-gap. In other words, such mid-gap levels can communicate with both extended bands, thus facilitating the possibility of creating a net positive or negative space charge.

Operationally, the detector is usually exposed to the X-ray radiation on the cathode side, so it can rely on the good transport properties of the electrons. The suppression of the electric field would be expected to be more pronounced and pushed towards the anode, similar to optical irradiation from the cathode side. However, since X-ray penetration lengths are tens to hundreds of microns, much longer than those of optical photons, a different behavior is actually achieved, with a pinch-off effect of the electric field well inside the detector, at a depth that increases with X-ray energy [33]. Whatever the source of the perturbation, the local collapse of the electric field inside the detector severely limits the charge collection, resulting in the radiation-induced polarization effect. As for the lateral perturbation of the electric field, it is mainly caused by the lateral diffusion of the photogenerated holes before being trapped. This is a cumulative effect that increases the perturbed region until the diffusion mechanism weakens. Diffusion is never dominant in the case of anode irradiation because the holes never experience negligible electric fields as they move from the anode to the cathode. In contrast, when the optical irradiation is focused on the cathode side [11], the electric field is nullified in a large circular region around the irradiated cathode, where ambipolar diffusion becomes dominant. Another difference is that the horizontal component of the field is directed either outward or inward in the case of anode or cathode irradiation, respectively.

In conclusion, we have seen that our experimental electro-optical imaging approach, together with appropriate data analysis, provides access to non-uniform x–y electric-field distributions and their time evolution under a voltage/optical bias sequence. We have studied the perturbation induced by the above-gap optical beam line focused on the planar semi-transparent anode contact. The largest perturbation is close to the cathode and appears in the lateral direction as a Gaussian distribution of positive space charge. Its width is larger than the 1-mm detector thickness in our experimental conditions.

The overall effect of the trapped holes can counterbalance the negative charge due to bias-induced polarization and improve the uniformity of the electric field during optical irradiation of the anode side.

The good agreement with the simulation confirms previous results obtained with the same model for cathode optical irradiation; this validates our two-level model and points to the central role of the deep compensation level in the electrical properties of CdTe Schottky detectors.

This imaging approach can be applied to detectors where the electric field is inherently non-uniform, such as those equipped with segmented electrodes such as strips or pixel 1D-arrays. In this case, it is possible to retrieve both (x and y) electric-field components when electrodes are continuous along the z-direction, i.e., that of the probe beam. With proper calibration, it could also be applied to the case of pixel 2D-arrays.

## Figures and Tables

**Figure 1 sensors-23-04795-f001:**
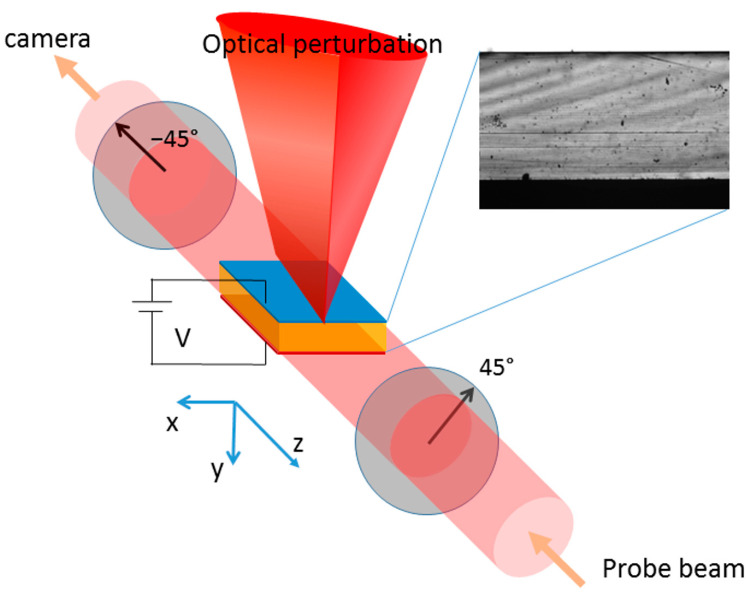
A sketch of the experiment: the line-focused optical excitation beam is incident on the anode side of the CdTe diode-like detector (**left**), whereas the 980 nm probe beam laterally crosses the detector in a cross-polarizer configuration. On the (**right**), an image of the transmitted intensity with both polarizers aligned and no voltage applied (P_para_).

**Figure 2 sensors-23-04795-f002:**
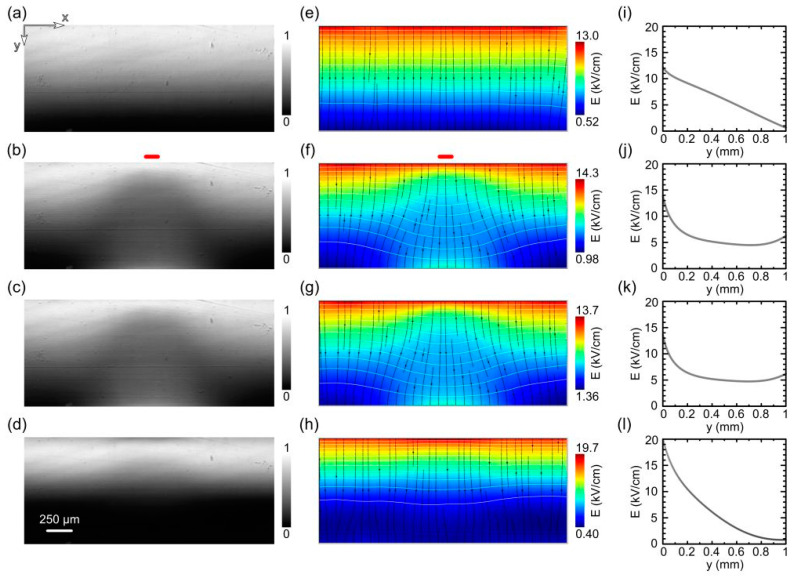
Normalized Pockels images (**left**), corresponding electric-field maps (**center**) and central vertical profiles of the electric field (from anode to cathode) at selected instants (**right**): (**a**,**e**,**i**) under dark, just before optical irradiation; (**b**,**f**,**j**) under light, after 5 min of optical irradiation (red bar denotes the position and size of the irradiated region); (**c**,**g**,**k**) under dark, 2 s after the light switch-off; (**d**,**h**,**l**) under dark, 15 min after the light switch-off. Isopotential lines (white lines) in 50 V steps are also reported in (**e**,**f**,**g**,**h**).

**Figure 3 sensors-23-04795-f003:**
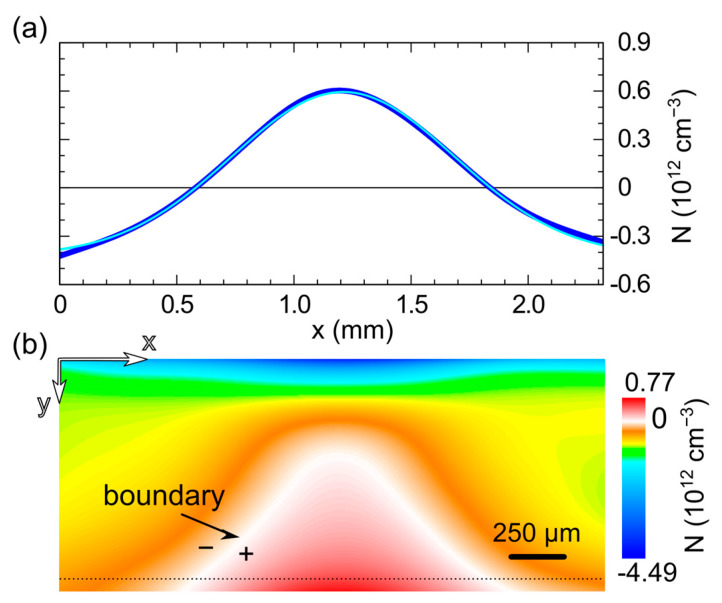
Experimental space charge: (**a**) the horizontal profile of the space charge (blue dots) and its fit by using a Gaussian function (cyan solid line). The profile is taken from the map below at a distance of 50 µm from the cathode. (**b**) Full map of the space charge at the end of the optical irradiation. The white color denotes values around zero, separating the regions bearing negative and positive charge. The electric-field data (Figure 2f) have been cleaned by using a fast Fourier transform filter before retrieving the charge. The anode is at the top side and the cathode below.

**Figure 4 sensors-23-04795-f004:**
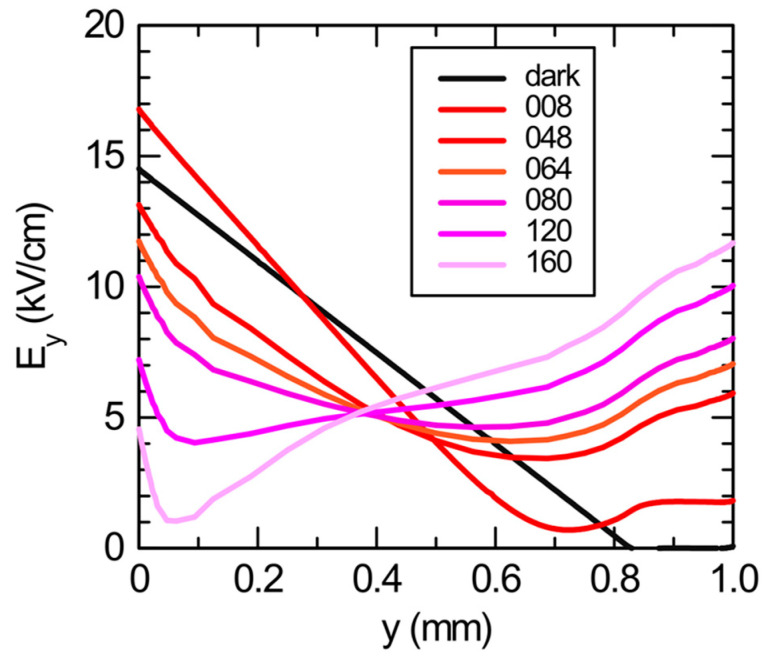
Simulated electric-field profiles at the center of the optical irradiation under different optical irradiances (in arb. units). Simulations refer to 5 min of optical irradiation. The profile under darkness is also shown (black line).

**Figure 5 sensors-23-04795-f005:**
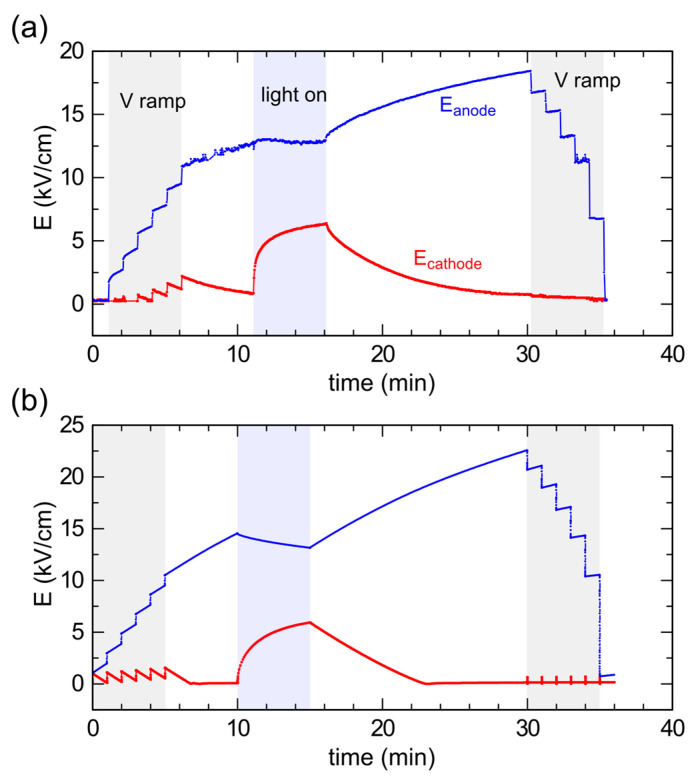
(**a**) Experimental and (**b**) simulated anode and cathode electric-field transients during the whole experiment (both fields are taken at 50 µm from the respective interface). The forward/backward voltage steps to/from 600 V and the optical irradiation interval are highlighted.

**Figure 6 sensors-23-04795-f006:**
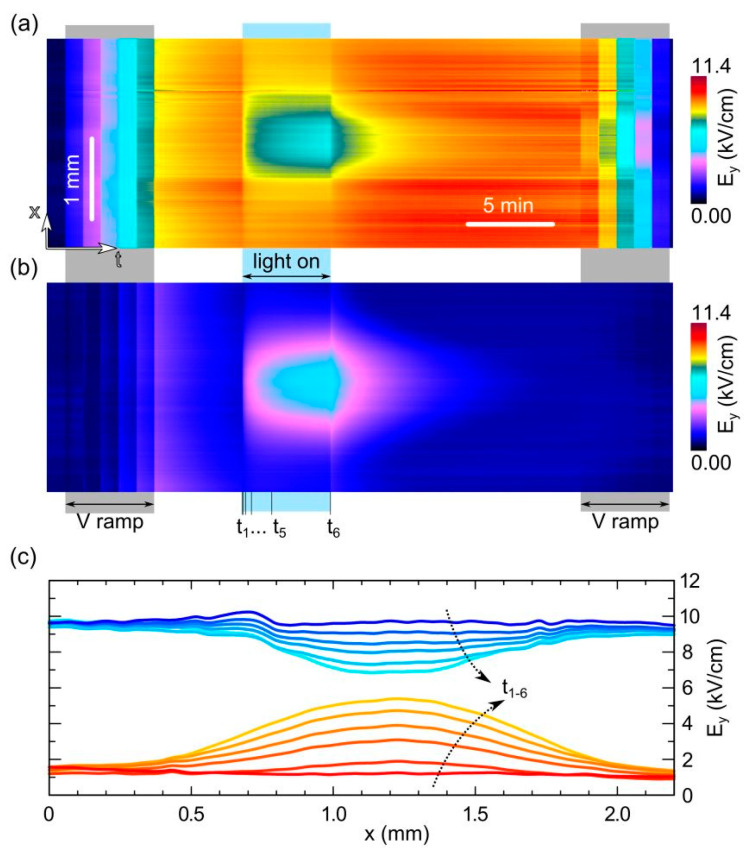
Time–space charts showing the horizontal crosscut of the electric field (vertical component) over time (**a**) near the anode and (**b**) near the cathode. The chart includes the V-ramp sequences up to 600 V and back to 0 V (in steps of 100 V). (**c**) Horizontal profiles of the electric field near the anode (upper bluish lines) and near the cathode (lower reddish lines). The first profile is in dark just before the light is turned on (darkest line) and the others are under light at different delays from the light switch-on: 2 s, 10 s, 30 s, 100 s, 300 s (t_1-6_ are also indicated in the panel above). The arrows indicate the increasing time. The field profiles are taken at 150 µm and 50 µm from the anode and cathode, respectively.

## Data Availability

The data presented in this study are available on request from the corresponding author.

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
