# Peer review of "Electric-Field Mapping of Optically Perturbed CdTe Radiation Detectors"

_sensors, 2023, doi:10.3390/s23104795_

Round 1
Reviewer 1 Report
It is considered suitable for publication.
Author Response
We thank the Referee for the positive evaluation
Reviewer 2 Report
The authors used an electro-optic imaging approach to evaluate how the localized free carriers photogenerated can affect the electric field distribution over a large spatial range and over a long time scale in CdTe Schottky detectors. The manuscript is interesting, but some questions should be answered:
The authors used the term “semi-insulating” which I understand as semi-conducting. Is it correct? If so, why did the authors used the “semi-insulating” term?
The authors changed the configuration by sending the perturbing beam on the anode side of the planar Schottky CdTe:Cl radiation detectors instead of the cathode side. It was not clear the reason why the authors change the electrode.
Authors should separate the value from the unity, like in “23mW” and “2sec”. Also, “sec” should be replaced by “s”.
The Results section is confused. It is hard to understand.
The field profile of the sample after 15 minutes in the dark after the irradiation, Fig. 2 l), does not present a linear behavior. It has an exponential decay profile, apparently. Could the authors discuss the result?
The authors obtained the FWHM from the horizontal profile of the space charge (Fig. 3). What is the relevance of this data? Do the authors could characterize the system with the gaussian fitting?
The inversion of the electric field profile is independent of the type of irradiated contact and bias sign, but why does it happen? Why is the hole trapping more effective near the cathode?
The manuscript is interesting. However, its presentation is poor. I believe the English and the clarity should be improved before publication so the paper will have the quality necessary for the readers.
Author Response
We thank the Referee for the careful evaluation. Hereafter, point-by-point responses are provided.
The authors used the term “semi-insulating” which I understand as semi-conducting. Is it correct? If so, why did the authors used the “semi-insulating” term?
Semi-insulating term refers to a material exhibiting an high electrical resistivity. It is commonly adopted when wide gap semi-conductors are doped/compensated such that Fermi levels lies at or near the middle of the energy gap thus resulting in electrical resistivity of the order or above 107ohmcm, which is the case for the CdTe crystals investigated in the present work.
The authors changed the configuration by sending the perturbing beam on the anode side of the planar Schottky CdTe:Cl radiation detectors instead of the cathode side. It was not clear the reason why the authors change the electrode.
The motives were already declared in the second part of the introduction. They are briefly recalled here: 1) to confirm the physics and its modeling, in particular to evaluate the role of photogenerated holes; 2) to confirm the experimental setup and the electric field recovery, in particular also when the device is irradiated from the more absorbing electrode; 3) to obtain more homogeneous electric field distributions under optical irradiation.
Authors should separate the value from the unity, like in “23mW” and “2sec”. Also, “sec” should be replaced by “s”.
We have corrected the text accordingly to the Referee’s comment.
The field profile of the sample after 15 minutes in the dark after the irradiation, Fig. 2 l), does not present a linear behavior. It has an exponential decay profile, apparently. Could the authors discuss the result?
This is an interesting feature which we have already addressed in the Results and Discussion sections.
In fact, the comment to panel of Fig.2l reads: “The last profile of Fig.2l shows that 15min after the optical irradiation, the field has almost retrieved its linear behavior as before the irradiation, but with a greater slope because some bias-induced-polarization has accumulated”.
One of the main effects of the illumination is to change the shape of the field from an essentially linear profile to a concave one. When removing the light, the profile only partially recovers the linear profile that is expected under dark, resulting in an intermediate shape. This is also confirmed by simulations.
We address such a “memory effect” in the Discussion also in terms of space charge. In particular, we observe that the space charge loses the space uniformity under irradiation and that the modified space charge rigidly shifts towards negative values after switching off the irradiation. This is the reason of the non-linear profile in Fig.2l. However, we have added a short comment to further clarify the concept (Text modification in red):
“Interestingly, the enhanced negative space charge near the anode continues to increase even after the optical excitation is removed, due to the hole emission process. In fact, since hole emission is the dominant process, the space charge rigidly shifts to more negative values throughout the active region. This phenomenology is evident when looking at the simulated profiles of the space charge in Fig.S2, which explains why the electric field profile does not recover the linear shape even 15 min after the light is turned off (Fig.2l)."
The authors obtained the FWHM from the horizontal profile of the space charge (Fig. 3). What is the relevance of this data? Do the authors could characterize the system with the gaussian fitting?
The horizontal profile of the space charge near the cathode exhibits a Gaussian profile. In the text we did not speculate about the origin of the profile, just used this evidence to quantify the extension of the perturbed region.
The inversion of the electric field profile is independent of the type of irradiated contact and bias sign, but why does it happen?
We agree with the Referee, this deserves a further comment. Initially, the field exhibits a decreasing linear profile, however the contact exposed to the optical irradiation undergoes a field suppression because of current continuity. The suppressed field must be recovered in the remaining detector depth, eventually resulting in the slope inversion. We have added a short comment to the text (in red):
“In agreement with Franc [32], we can state that at high optical irradiances, the influence of the contacts becomes less pronounced with respect to the suppression induced by the optical irradiation. In addition to device-dependent peculiarities, the electric field suppression under the optically irradiated contact is expected due to the continuity of the electric current."
Why is the hole trapping more effective near the cathode?
We have already addressed this point in the Discussion, however we further clarify this as shown in the following (modified text in red):
“During optical irradiation, or at least at its beginning, hole trapping near the cathode is more effective, because the holes experience lower and lower electric fields as they drift. This leads to an accumulation of positive space charge near the cathode, which is counterbalanced by the negative charge near the anode, resulting in the observed electric field convexity.”
The manuscript is interesting. However, its presentation is poor. I believe the English and the clarity should be improved before publication so the paper will have the quality necessary for the readers.
We have improved the English and clarity accordingly.
Reviewer 3 Report
The authors present an interesting study investigating the two-dimensional electric field in a Schottky CdTe detector using the Pockels effect. In the introduction, the authors demonstrate the importance of understanding the spatial distribution of electric fields in radiation detectors and how they are perturbed by incident radiation. Probing the non-equilibrium electric field distribution in CdTe Schottky detectors, such as those leading to polarization, could help improve the performance of planar or electrode-segmented detectors. To measure the distribution of the electric field, the set-up utilizes a perturbing optical beam on the anode side of the CdTe detector with a probe beam to measure the transmission. The measurement results are in agreement with numerical simulations, confirming a 2-level model based on a dominant deep level, which fully accounted for the temporal and spatial dynamics of the perturbed electric field. These findings are particularly relevant for improving the accuracy and reliability of radiation detectors, with implications for many fields, including medical imaging and nuclear physics. Overall, the manuscript is well-written, and the results are clearly presented. Thus, I recommend this manuscript for publication.
Author Response
We thank the Referee for his/her valuable comments.
Reviewer 4 Report
Please find below my comments about the manuscript titled “Electric Field Mapping of Optically Perturbed CdTe Radiation Detectors''.
 General comments
The authors present the results from electric field mapping in CdTe detectors. Currently, CdTe and CdZnTe detectors represent the best products for room temperature X-ray and gamma ray detection. Of course, the spatial distribution of the electric field plays a key role in the development of radiation detectors. This work is interesting to read and the topic is very appealing.
To strengthen the paper, some minor remarks are reported below.
Minor Remarks
Line 45
Other research groups studied the electric field profiles in CdTe detectors. To enhance the introduction, it would be interesting to cite the following references:
[1] A.A. Turturici et al., Time-dependent electric field in Al/CdTe/Pt detectors. Nucl. Instr. Meth. A 2015, 795, 58-64.
[2] A.A. Turturici et al., Electric field manipulation in Al/CdTe/Pt detectors under optical perturbations. Nucl. Instr. Meth. A 2017, 858, 36-43.
Line 67
The authors correctly stressed the critical issues concerning the charge losses at the inter-electrode gap, due to charge sharing events in segmented detectors. These phenomena are related to the presence of distortions of the electric field profile near the gap. Is your detector characterized by the presence of a guard-ring on the electrodes? If yes, it would be interesting to investigate the electric field profile near the gap between the pixel and the guard-ring.
Best Regards
Author Response
We thank the Referee for his/her valuable comments and the positive evaluation. Hereafter we address his/her points.
Minor Remarks - Line 45
Other research groups studied the electric field profiles in CdTe detectors. To enhance the introduction, it would be interesting to cite the following references:
[1] A.A. Turturici et al., Time-dependent electric field in Al/CdTe/Pt detectors. Nucl. Instr. Meth. A 2015, 795, 58-64.
[2] A.A. Turturici et al., Electric field manipulation in Al/CdTe/Pt detectors under optical perturbations. Nucl. Instr. Meth. A 2017, 858, 36-43.
We agree with the Referee and added the suggested references.
Line 67
The authors correctly stressed the critical issues concerning the charge losses at the inter-electrode gap, due to charge sharing events in segmented detectors. These phenomena are related to the presence of distortions of the electric field profile near the gap. Is your detector characterized by the presence of a guard-ring on the electrodes? If yes, it would be interesting to investigate the electric field profile near the gap between the pixel and the guard-ring.
No, our detector is fully planar, without guard-ring. The Referee is right: our procedure would allow to evaluate the electric field laterally, between the pixel and the guard-ring in the case of such devices. However, some corrections would be necessary to account for the presence of the guard-ring segment perpendicular to the optical path.